# Carbon Reduction and Pollutant Abatement by a Bio–Ecological Combined Process for Rural Sewage

**DOI:** 10.3390/ijerph20021643

**Published:** 2023-01-16

**Authors:** Qiu Jin, Liangang Chen, Shengyun Yang, Chaochao Zhu, Jingang Li, Jing Chen, Wei Li, Xinxin Peng

**Affiliations:** 1State Key Laboratory of Hydrology-Water Resources and Hydraulic Engineering, Nanjing Hydraulic Research Institute, Nanjing 210029, China; 2School of Agricultural Science and Engineering, Hohai University, Nanjing 210098, China; 3Department of Ecological Sciences and Engineering, College of Environment and Ecology, Chongqing University, Chongqing 400030, China

**Keywords:** wastewater treatment, seasonal changes, microbial community, constructed wetlands

## Abstract

In order to explore the treatment effect of a bio–ecological combined process on pollution reduction and carbon abatement of rural domestic wastewater under seasonal changes, the rural area of Lingui District, Guilin City, Guangxi Province, China was selected to construct a combined process of regulating a pond, biological filter, subsurface flow constructed wetland, and ecological purification pond. The influent water, effluent water, and the characteristics of pollutant treatment in each unit were investigated. The results showed that the average removal rates of COD, TN, and NH_3_–N in summer were 87.57, 72.18, and 80.98%, respectively, while they were 77.46, 57.52, and 64.48% in winter. There were significant seasonal differences in wastewater treatment results in Guilin. Meanwhile, in view of the low carbon:nitrogen ratio in the influent and the poor decontamination effect, the method of adding additional carbon sources such as sludge fermentation and rice straw is proposed to strengthen resource utilization and achieve carbon reduction and emission reduction. The treatment effect of ecological units, especially constructed wetland units, had a high contribution rate of TN treatment, but it was greatly impacted by seasons. The analysis of the relative abundance of the microbial community at the phylum level in constructed wetlands revealed that Proteobacteria, Acidobacteria, Chloroflexi, Firmicutes, Bacteroidetes, Planctomycetota, and Actinobacteria were the dominant phyla. The relative abundance of microbial communities of Proteobacteria, Chloroflexi, and Acidobacteria decreased to a large extent from summer to winter, while Firmicutes, Bacteroidetes, and Planctomycetota increased to varying degrees. These dominant bacteria played an important role in the degradation of pollutants such as COD, NH_3_–N, and TN in wetland systems.

## 1. Introduction

Rural domestic wastewater has the characteristics of scattered pollution sources, large total amount, small treatment scale, low treatment rate, high wastewater discharge load, and large fluctuation of water quantity and quality [1]. Restricted by decentralized residential mode, geographical environment, and economic conditions, most rural wastewater is difficult to connect to urban drainage pipe network systems. Wastewater treatment technology applied to cities is hard to popularize in rural areas, causing high difficulty in wastewater treatment. In addition, there are still some problems in rural wastewater, including large fluctuation of discharge, difficulty in the centralized collection, lack of funds and technologies that can be used to maintain the long-term operation of wastewater treatment facilities, weak environmental awareness of rural residents, and shortage of talents with scientific literacy and technology. Thus, the rural wastewater treatment model with low consumption, high efficiency, ecology, and easy management is crucial.

The bio–ecological collaborative treatment model is to combine the biological treatment method and the ecological treatment method of wastewater, and fully combine the microbial degradation capacity in the biological treatment stage with the fixed absorption capacity of carbon and nitrogen in the ecological treatment stage, so as to achieve the purpose of advanced treatment of wastewater [2]. In recent years, research on the application of rural wastewater focuses on wastewater treatment technology and the treatment effect of wastewater quality indicators. Some common treatment technologies, such as anaerobic baffled reactor (ABR), often have problems such as the limitation of treatment effect, long hydraulic retention time, and large occupied area when treating the actual low-concentration rural wastewater. It has been reported that biological carrier composite membrane bioreactors and suspended carrier composite membrane bioreactors can effectively remove COD, NH_3_–N, and TN in the treatment of low-C/N-ratio wastewater, but the high cost made them unsuitable for rural wastewater treatment [3]. In addition, the treatment effect of the bio–ecological combined system, especially the ecological unit, is affected by seasonal changes in the outdoor environment, and the microbial community and structure also changed, which influenced both the microbial metabolism process and the removal efficiency of pollutants [4].

In this study, the rural area of Lingui District, Guilin City, Guangxi Province, China was selected as the study site. There are 40 households in the village with a population of 168. Based on in-depth analysis of the characteristics of water quality and quantity of rural domestic sewage in Guangxi and the advantages and disadvantages of biological and ecological treatment technologies, the bio–ecological combination process of the anaerobic pool, biological filter, artificial wetland, and step-by-step biological-control, ecological-purification pond was adopted. The COD and TN removal rates were taken as the target analytes due to their large number and high representativeness, and the target concentrations of each unit were collected in the summer and winter of 2021. This study aimed to analyze the seasonal features of wastewater treatment, discuss the contribution of the ecological units to the pollutant removal rate, and further track the changes in the microbial community structure in the wetland system. These findings can provide a reference for improving and promoting bio–ecological treatment technology in rural domestic wastewater treatment in China, as well as a scientific basis for wastewater treatment in rural areas of Guangxi, China.

## 2. Material and Methods

### 2.1. Experimental Device

The bio–ecological combined process used in this study was mainly composed of an anaerobic tank, biological filter, constructed wetland, oxidation pond, and some supporting facilities. The area of the whole wastewater treatment facility was about 160 m^2^, and the design water inflow was 12 m^3^/d (Figure 1).

The volume of the anaerobic tank was 2.8 m × 1.7 m × 1.8 m, and a submersible pump was equipped at the rear of the anaerobic reaction tank. The biofilter was trapezoidal on the plane with a 2.9 m upper bottom, 3.1 m lower bottom, and 1.6 m long along the water flow direction (excluding the cell body). The depth of the biofilter was 1.35 m, and the effective water depth was 1.1 m. In the experiment, the submersible pump switch was controlled by the float value and was started when the water level reached a certain level to lift the wastewater into the biological filter. The biofilter was filled with 40 cm ceramsite (particle size of 5–8 mm) and 40 cm porous media (particle size of 150 mm) from bottom to top. The ceramsite layer and the porous medium layer were separated by a 3-centimeter-thick grid. The filter tank used the methods of water drop and air extraction pipe to supplement the dissolved oxygen. The submersible wastewater pump pumped water to a high place, dropped water through the water distributor, and dropped water through the inlet pan. The built-in air extraction pipe was used for supplying the dissolved oxygen, so that the oxygen could be charged during the process of falling and splashing, and the water could be evenly distributed. According to our long-term research on the oxygenation effect of the biofilter, the method applied to the device that used nature ventilation and splashing tray to strengthen oxygenation could satisfy the oxygen demand of microorganisms in the biofilter. The subsurface flow constructed wetland was trapezoidal on the plane with an upper bottom of 3.1 m and the lower bottom of 9.6 m, and 11.2 m long and 1.35 m deep along the water flow direction. The front end of the wetland was a 0.8-meter-long inlet channel. The wetland fillers were stone slag (20 cm), 30% stone slag mixed with 70% backfilling soil (30 cm), and powdery cinder (30 cm) from bottom to top. The height of the filler was 80 cm, belonging to the subsurface flow constructed wetland. Wild rice stem and canna were mixed planted in the wetland, with a planting density of 1 dm^2^/plant. The hydraulic gradient of the wetland was close to 1%, and the effective water depth was 90 cm. There were two DN50 water outlets in the upper part of the wetland, and the height from the bottom of the pool was 0.9 m. The oxidation pond was trapezoidal on the plane with an upper bottom of 9.6 m, lower bottom of 12.7 m, and 7.2 m long along the water flow direction (excluding the pool body). The depth of the oxidation pond was 1.35 m, and the thickness of the pond was 0.2 m. A stainless-steel mesh (3 mm hole) was used in the oxidation pond and the pipes were arranged to stabilize the mesh. The pond was divided into four areas along the flow direction. Zooplankton resting eggs, zoobenthos (pond snails, mussels, chironomid larvae), fish (silver carp, bighead carp), and aquatic plant (water celery, water spinach) juveniles were added to each compartment, respectively, to form the ecological purification function areas containing the zooplankton filter feeding area, zoobenthos scraping feeding area, fish feeding area, and aquatic plant water stability area.

The domestic wastewater in the study area was collected into the anaerobic tank through the pipe network, where organic matter was degraded or partially degraded and complex macromolecules were decomposed into small molecules with simple structures. Meanwhile, it played a role in water quantity adjustment and quality balance. The submersible pump was at the rear of the anaerobic tank. The submersible wastewater pump pumped water to a high place, dropped water through the water distributor, and dropped water through the inlet pan, with a built-in air extraction pipe for oxygen charging. The effluent from the filter tank entered the wetland inlet tank by bottom effluent and then entered the back end of the wetland through the overflow dam. The removal effect of N and P was enhanced by the joint action of microorganisms, plants, and fillers in the wetland. The effluent from the wetland entered the oxidation pond, where the remaining pollutants were deeply treated by the ecological purification function areas, including the zooplankton filter feeding area (area 1), zoobenthos scraping feeding area (area 2), fish feeding area (area 3), and aquatic plant water stability area (area 4), to realize wastewater recycling.

### 2.2. Operational Condition

In this study, the domestic wastewater in Da Antou Village, Lingui District, Guilin city was selected as the raw water. The wastewater quantity was greatly affected by seasonal changes. The actual water intake in summer was 5–6 m^3^/d, while in winter it became 3–5 m^3^/d due to the significant descent in residential water consumption. The influent water quality in summer and winter is shown in Table 1. The hydraulic load and fixed hydraulic retention time (HRT) of each unit are shown in Table 2.

### 2.3. Monitoring and Data Acquisition

In this study, the water samples were collected from 18 July 2021, in the summer, and from 26 November 2021, in winter. The samples were taken every two to three days to monitor the water quality. Four sampling points were selected, including the inlet of the biofilter, the inlet of the constructed wetland (the front end of the inlet flume), the outlet at the end of the wetland, and the outlet at the end of the oxidation pond. Water quality monitoring items and specific methods are shown in Table 3. During the sampling period, three sampling points were at the front, middle, and back of the constructed wetland to collect the water samples in the canna planting area, wild rice stem planting area, and canna planting area, respectively. DNA extraction and Miseq high-throughput sequencing of the microbial filter membrane samples were entrusted to Bioinformatics Technology Co., Ltd. (Shanghai, China) to analyze the structures and differences in microbial communities at each sampling point in the wetland system.

## 3. Results and Discussion

### 3.1. Water Quality Index and Wastewater Treatment Effect

#### 3.1.1. The Removal Effect of Combined Process in Summer

During the summer experiment period (mainly from July to August), the air temperature and water temperature were 26–34 °C and 25–30 °C, respectively. The pH and DO were 6.8–8.0 and 2.9–6.1 mg/L, respectively. The overall purification effect of the summer combined system on the three wastewater indicators was significant (Figure 2). The average removal rates of COD, NH_3_–N, and TN were 87.57, 80.98, and 72.18%, respectively. The mass concentration of COD, NH_3_–N, and TN in influent were 101.83–332.30, 8.94–44.75, and 21.28–79.38 mg/L, respectively. Different from the fluctuation of the influent concentration, the effluent concentration was stable at a low state. The average effluent concentration reached the first-class A discharged standard of “Discharge standard of pollutants for municipal wastewater treatment plant” (GB 18918-2002) (the mass concentrations of COD, NH_3_–N, and TN in the effluent are not higher than 50, 5, and 15 mg/L). The effluent quality was stable, since the combined process of the regulating tank, biofilter, constructed wetland, and ecological purification pond could effectively buffer, regulate, and degrade wastewater.

After the pretreatment of the collected rural domestic wastewater by the regulating tank, the refractory organic matter in the water was decomposed into simple organic matter, which greatly improved the biodegradability of the wastewater. The wastewater was pumped into the biofilter through the submersible pump, fully contacted with air by the graded drop, and further oxygenated by the air extraction pipe to achieve an efficient oxygen supply. In the biofilter, the COD was highly removed by microbial assimilation reaction. In the ecological section (constructed wetland and oxidation pond), the residual organic matter in wastewater was further removed by microbial utilization, substrate adsorption, and plant filtration [5]. The treatment effect of COD was strengthened through the combination of biological and ecological sections. After the treatment of the combined process, the average effluent concentration of COD was decreased to 21.66 mg/L, far lower than the first-class A discharge standard of GB 18918-2002.

For the TN removal, the combined process overcame the problems of the limited nitrogen removal effect of traditional biological treatment and unstable removal effect of ecological treatment, and obtained a relatively stable nitrogen removal effect. The average effluent concentration of TN was 10.19 mg/L, meeting the first-class A discharge standard of GB 18918-2002. The biofilter mainly removed nitrogen by biological denitrification (nitrification and denitrification) and microbial assimilation of some nitrogen in wastewater. The constructed wetland mainly supplied oxygen from plants to the inside. Aerobic, anoxic, and anaerobic zones were formed near the roots according to the content of dissolved oxygen, and provided conditions for the microbial nitrification–denitrification reaction. In addition, some inorganic nitrogen (ammonia nitrogen, nitrate nitrogen, etc.) could be removed by plant absorption [6]. The oxidation pond (ecological purification pond) could reduce nitrogen by organic nitrogen deposition, biological nitrification and denitrification, and aquatic plant absorption. However, the predominant mechanism of nitrogen removal was still controversial [7].

In the biofilter, the microbial attachment growth method was beneficial for the culture of nitrification bacteria. Appropriate temperature and sufficient oxygen supply also provided conditions for NH_3_-N to be oxidized to nitrate or nitrite by nitrification. The NH_3_-N removal in constructed wetlands mainly depended on the nitrification reaction in the aerobic zone near the plant roots. Additionally, a part of NH_3_-N was reduced by plant absorption, packing adsorption, and volatilization. There were still many ways to remove NH_3_-N in the oxidation pond, including biological nitrification, aquatic plant absorption, and NH_3_-N stripping [7]. Through the three levels of guarantee, the average effluent concentration in this process was low (2.84 mg/L), and could meet the discharge standard.

#### 3.1.2. The Removal Effect of Combined Process in Winter

During the winter experiment period (mainly from November to December), the air temperature and water temperature were 8–20 °C and 0–10 °C, respectively. The pH and DO were 6.9–8.2 and 2.4–5.5 mg/L, respectively. The removal rates of COD, NH_3_–N, and TN in this combined system in winter were 74.54% (60.49–85.21%), 64.48% (55.20–81.94%) and 57.52% (48.82–72.22%), respectively, and the average effluent concentrations were 37.16, 4.50, and 33.04 mg/L, respectively (Figure 3). The average effluent concentrations of COD and NH_3_–N reached the first-class A discharged standard of GB 18918-2002. The overall removal effect was high, but significantly lower than the effect in summer.

The treatment effect of nitrogen was greatly affected by seasonal changes, and did not meet the first-class A discharged standard of GB 18918-2002. This may be because the low temperature affected the microbial activity of the combined system and weakened the nitrification and denitrification of the biological–ecological process, which influenced the migration and transformation of nitrogen. Moreover, impacted by seasonal changes, the growth of plants was limited in winter, which influenced the nitrogen removal effect of constructed wetlands relying on plant root absorption [8]. The roots of wild rice stems and canna were well-developed in the wetland. The distribution of nitrification bacteria in roots to the deep wetland provided an aerobic micro-environment, which benefited the nitrification reaction. However, the wetland with wild rice stem and canna was greatly affected by temperature. The absorption ability of inorganic nitrogen was much lower in winter than in summer, which may have led to seasonal differences in the nitrogen removal effect. The temperature changes had more influence on the removal effect of TN and NH_3_-N than COD. This may be because the temperature coefficient of assimilation was lower than nitrification and denitrification [9].

### 3.2. Analysis of Ecological Unit Treatment Effect

#### 3.2.1. Contribution Rate of Pollutant Removal in Ecological Units

The ecological units in the system included constructed wetlands and oxidation ponds. The contribution rates of the ecological units to pollutant removal in summer and winter are shown in Figure 4. The contribution rate of ecological units to TN removal was significantly higher than that of biological units in both summer and winter, and the contribution of constructed wetland to TN removal was the largest (65.08% in summer and 53.15% in winter), which was mainly due to the alternation of aerobic, anoxic, and anaerobic environments in the wetland. In this environment, denitrifying bacteria could play a full role, and plants in the wetland could also absorb part of the inorganic nitrogen.

The low contribution rate of the oxidation pond to TN removal may be due to the low carbon and nitrogen content of the influent water (C/N ratio between 2:1 and 3:1) and the serious shortage of carbon source required by microorganisms, which greatly weakened the denitrification to a certain extent and affected the denitrification effect of the oxidation pond. The contribution rate of oxidation pond to COD removal was the lowest (6.46% in summer and 12.24% in winter), which may be due to the small load of organic matter in the influent water, resulting in the reduction in nutrients available for microbial growth, thereby reducing the microbial activity. Meanwhile, some algae were trapped in the oxidation pond, and their decay and decomposition may cause a secondary pollution of organic matter [10].

The purification effect of the ecological unit was relatively low in winter. During the stage of wilting and decaying, the adsorption and oxygen supply capacity of plant roots decreased. The purification capacity of ecological units, especially constructed wetlands, was reduced. Pollutants that had been absorbed in the plants returned to the water. The low temperature also reduced microbial activity. In addition, the flow velocity decreased in winter. The high concentration of pollutants, even beyond the system load, was also a major factor affecting the purification capacity of the system in winter.

#### 3.2.2. Contribution Rate of Pollutant Removal in Ecological Units

In order to further analyze the difference in the removal effect of ecological units, especially the wetland system under seasonal changes, and study the difference in the composition and structure of the microbial community in the wetland system, the relative abundance at phylum level of the microbial community in the wetland system was obtained through sequencing analysis. In order to facilitate the analysis of microbial community at all levels of wetlands, phyla with relative abundance higher than 2% were selected as the main phyla.

There was no difference in the species of dominant phyla in each sampling site of the wetland at the phylum level, while there was a little difference in relative abundance (Figure 5). The main dominant bacteria were Proteobacteria, Acidobacteria, Chloroflexi, Firmicutes, Bacteroidetes, Planctomycetota, and Actinobacteria. The total sequence of these seven types of microorganisms accounted for more than 80% of the total sequence number. Relevant studies have shown that Proteobacteria was a group with a high content in the wetland bacterial community, and was the dominant bacterial group in the wetland system. It played an important role in the removal of TN and the degradation of carbohydrates and other organic compounds in the wetland system [11]. Most Actinobacteria belonged to heterotrophic bacteria, which could decompose much organic matter and participated in the degradation process of complex organic matter [12]. Acidobacteria, Chloroflexi, and Bacteroidetes were mainly dominant bacteria in the filler and sediment, and were conducive to the mineralization and degradation of organic matter in sewage [13]. The total number of Acidobacteria in the microbial community was less than 5%, which adapted to the oligotrophic environment and contributed to the carbon and nitrogen cycle [14]. Acidobacteria grew slowly because of the low energy generated by bacterial metabolism, and its low growth rate may make it difficult for them to compete with other groups in the community. The phylum Chloroflexi could effectively remove organic carbon by using carbon cycling, decomposition of organic compounds, fixation of CO_2_, production of acetic acid (volatile fatty acids and acetate), and the formation of ATP through substrate level phosphorylation [15]. Bacteroidetes and Firmicutes played an important role in the denitrification process [16].

The relative abundance of Proteobacteria, Chloroflexi, and Acidobacteria decreased to a large extent from summer to winter, while that of Firmicutes, Bacteroidetes, and Planctomycetota increased to varying degrees. The relative abundance of Actinobacteria changed little from summer to winter.

In general, the Proteobacteria was obligate or facultative anaerobic metabolism, including nitrification bacteria and the vast majority of denitrifying bacteria genera, which played an important role in wetland nitrogen removal [17]. The reproductive rate and metabolic activity of nitrifying bacteria and denitrifying bacteria were limited by low temperatures (less than 15 °C), and their relative abundance might decrease significantly with a decrease in temperature. This may be the reason for the significant seasonal variation in the nitrogen removal effect in wetlands. In winter, plants became dormant or died due to lack of oxygen, and the water transparency decreased. Chloroflexi was a kind of photoautotrophic bacteria involved in the transformation of inorganic carbon [18]. It was mainly distributed in the anoxic area of water irradiated by light energy. In winter, the light permeability of wetlands deteriorated, and the relative abundance of Chloroflexi may decrease. Acidobacteria was a rhizosphere bacterium, which accounted for the majority of aerobic bacteria and was greatly influenced by plants. They could participate in the biogeochemical cycle of elements such as carbon and nitrogen in plant roots [19]. The wilt and collapse of aquatic plants led to a significant decline in the relative abundance of aerobic and facultative anaerobic microorganisms in constructed wetlands. The relative abundance of Firmicutes was positively correlated with the nutrient level of sediments and the mud–water interface [20], which may be affected by the branches and leaves falling into the water in winter. The content of carbon, nitrogen, and organic matter in the wetland was relatively high, so the relative abundance of Firmicutes in winter was higher than that in summer. The relative abundance of Planctomycetota changed significantly with seasonal changes. The microorganisms in Planctomycetota contained anaerobic ammoniating bacteria as well as some other aerobic bacteria, so further screening and analysis were needed. Bacteroidetes could degrade macromolecular carbohydrates and participate in the carbon cycle of litter [21]. As Bacteroidetes could adapt to seasonal fluctuations, the relative abundance of Bacteroidetes increased in winter.

### 3.3. Analysis of C/N of Sewage

Among the 30 sampling measurements in this study, 27 water samples had C/N < 8 (Figure 6), and therefore they belonged to low-carbon:nitrogen-ratio sewage (sewage with COD/TN < 8 or BOD/TN < 5 is generally defined as low-carbon:nitrogen-ratio sewage [22,23]). Low-C/N wastewater treatment was often accompanied by high TN in the effluent. The average TN removal rate of this system was 72.18% in summer and 57.52% in winter, which is because there were not many carbon sources in the wastewater, leading to few electron donors for the denitrification of denitrifying bacteria, and denitrification was inhibited, resulting in a low TN removal rate.

To improve the efficiency of nitrogen removal from low-C/N wastewater, the technology of external carbon sources and non-external carbon sources are often used. The commonly used supplementary carbon sources are glucose, methanol, acetic acid, sodium acetate, etc. These carbon sources are easy to degrade and had high nitrogen removal efficiency and low sludge yield. However, they have high costs and strict transportation conditions. In addition, some organic compounds have certain biological toxicity. In order to treat wastewater at a low cost and with energy conservation and emission reduction, sludge ferment and natural cellulose have become popular choices. The sludge fermentation uses the residual activated sludge anaerobic fermentation to produce short-chain fatty acids such as acetic acid, propionic acid, and butyric acid as additional carbon sources. On the one hand, it can provide a cheap carbon source for treating low-C/N wastewater, and strengthen the removal of nitrogen and phosphorus in wastewater. On the other hand, residual sludge is reduced and recycled [24]. Yuan et al. used sludge fermentation as an additional carbon source for domestic sewage with a low C/N. The results showed that the introduction of fermentation did not affect the stability of the bacterial structure in the wastewater nitrogen and phosphorus removal system, but could significantly improve the efficiency of nitrogen removal [25]. Yao et al. used sludge fermentation liquid as an additional carbon source to enhance the performance of nitrogen and phosphorus removal in the micro-aeration ditch, and found that the stabilized effluent could all meet the first-class A standard stipulated by China (COD ≤ 50 mg/L, TN ≤ 15 mg/L, NH_3_-N ≤ 5 mg/L) [26]. Natural cellulose carbon sources with rich sources, and low-cost, slow-release carbon sources such as straw, willow, corn cobs, and reed rods, are widely used in artificial wetland units. Xiao et al. used willow dishes as a carbon source in vertical flow constructed wetlands, and found that the denitrification effect greatly ascends after adding the carbon source, and the maximum removal rate of nitrate nitrogen was 91.2% [27]. Zhao et al. added reed rods to the vertical flow constructed wetland, and found that the TN removal rate increased to 80% when the reed rods were added at 1.0 kg/m [28]. Zhang et al. added corncob into the horizontal subsurface flow constructed wetland, and the results showed that the removal rates of TN and TP increased by 35.82% and 9.70%, respectively [29].

In general, rice straw, corn cobs, and common agricultural waste composting have been commonly added to the ecological unit (wetlands) in Guangxi, which has increased sewage carbon source, promoted microbial nitrification and denitrification, and improved the efficiency of nitrogen. Meanwhile, agricultural waste resources are recycled and the power consumption cost of sewage treatment is reduced, which can effectively reduce carbon emissions.

## 4. Conclusions

During the operation of the combined equipment of the regulating tank, biological filter tank, subsurface flow constructed wetland, and ecological purification pond, although the quality of the sewage inflow fluctuates greatly, the application of this process in rural areas of Guangxi, China still achieves good results, and the aquatic vegetables, flowers, hygrophytes, fish, and benthic animals produced by the device can produce certain economic benefits. Through the contribution rate of each unit of the combined process to the removal of pollutants, different unit combinations can be designed in rural areas according to local conditions, effectively and at low cost, to solve the problem of rural domestic sewage dumping at will, which makes the environment worse and increases the pressure of carbon reduction. The effluent is connected to the farmland and orchard for irrigation, realizing resource utilization and fixing the carbon in the sewage. In addition, considering the low C/N of the influent in this system, common agricultural wastes such as sludge ferment and straw can be appropriately added to the biofilter and constructed wetland to improve the sewage purification effect, save costs, realize resource utilization, reduce carbon, and reduce emissions. The results of further microbial structure distribution showed that the main dominant bacteria in the constructed wetland were the same in summer and winter, and the relative abundance of microorganisms changed little from summer to winter. Therefore, the carbon reduction effect of the combined process is less affected by the season. As a carbon reduction and emission reduction sewage treatment technology in relatively mild areas, it is very much worth promoting.

## Figures and Tables

**Figure 1 ijerph-20-01643-f001:**
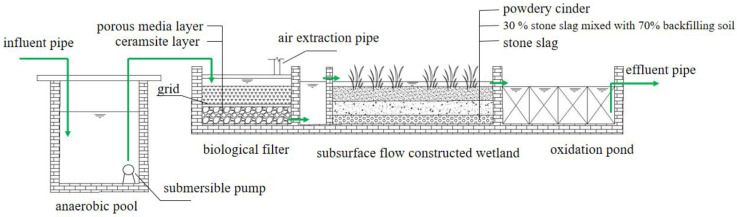
Schematic diagram of the apparatus.

**Figure 2 ijerph-20-01643-f002:**
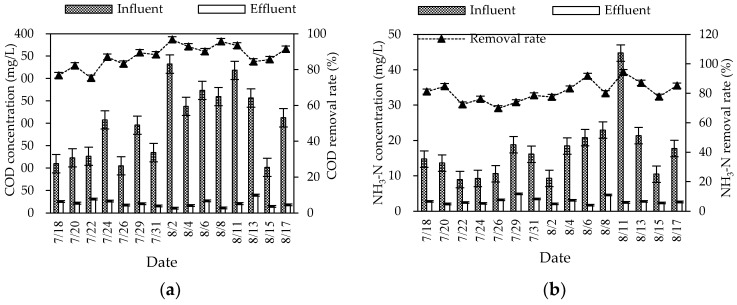
Changes in summer COD (**a**), NH_3_–N (**b**), and TN (**c**) concentrations and removal rates in the combined system.

**Figure 3 ijerph-20-01643-f003:**
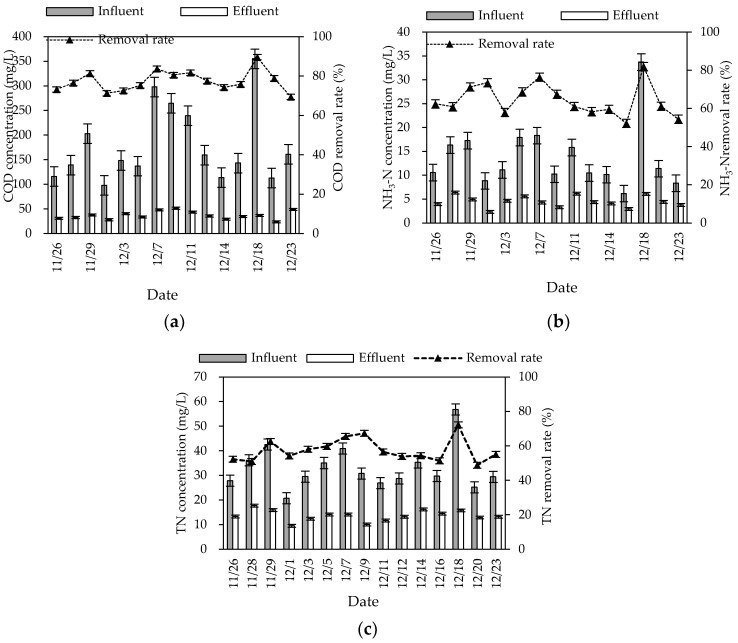
Winter inlet and outlet COD (**a**), NH_3_–N (**b**), and TN (**c**) concentrations and removal rates of the combined system.

**Figure 4 ijerph-20-01643-f004:**
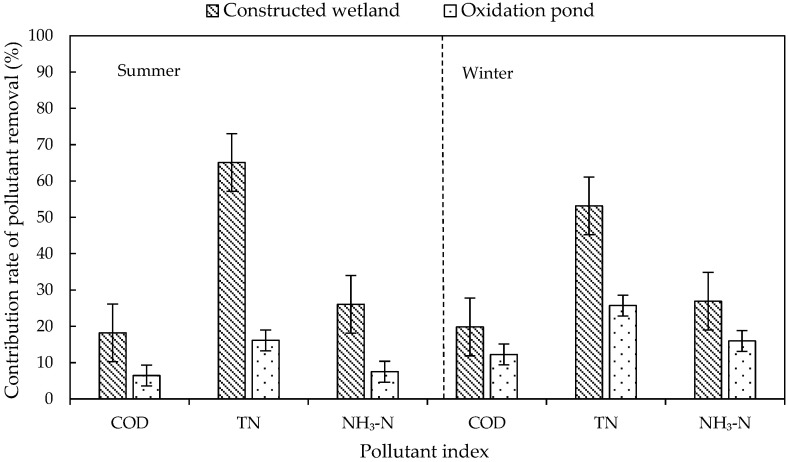
Variation in contribution rate of pollutant removal rate in ecological units.

**Figure 5 ijerph-20-01643-f005:**
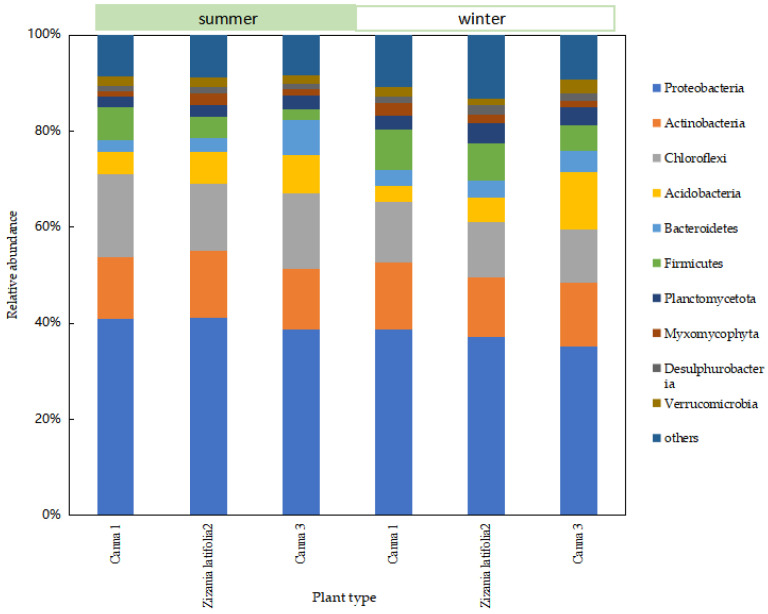
Relative abundance of microbial communities at the phylum level of wetland systems.

**Figure 6 ijerph-20-01643-f006:**
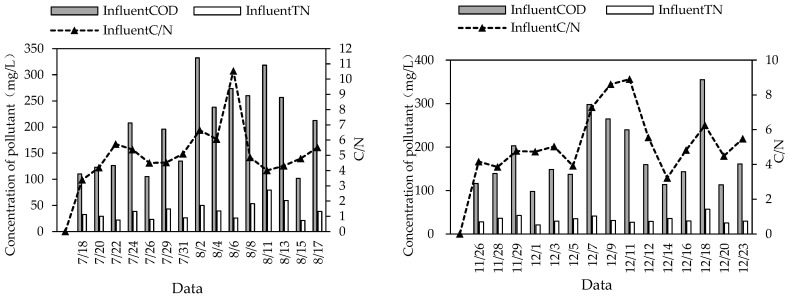
C/N of influent water in combined system in summer and winter.

**Table 1 ijerph-20-01643-t001:** The quality of the inlet water mg/L.

Item	*ρ*(COD)	*ρ*(NH_3_–N)	*ρ*(TN)
Measured value in summer	101.83–332.30	8.94–44.75	21.28–79.38
Average value in summer	199.79	17.19	38.86
Measured value in winter	98.01–355.12	6.17–33.72	20.72–56.80
Average value in winter	179.28	13.77	33.04

**Table 2 ijerph-20-01643-t002:** Hydraulic load and fixed hydraulic residence time of each unit.

Parameter		Anaerobic Pool	Biological Filter	Constructed Wetland	Oxidation Pond
**Hydraulic load m^3^/(m^2^·d)**	Summer	1.26	1.25	0.08	0.07
	Winter	1.05	1.04	0.07	0.06
**HRT (d)**	Summer	0.875	0.875	11	11
	Winter	1.042	1	13	113

**Table 3 ijerph-20-01643-t003:** Water quality monitoring projects and methods.

Monitoring Project	Monitoring Method	Instrument
Chemical oxygen demand (COD)	Potassium dichromate method/ Fast catalytic oxidation method	COD determination instrument Model CTL-12A
DO	Oxygen dissolved method	Portable dissolved oxygen meter Model WTWoxi330 in German
Total nitrogen (TN)	K_2_S_2_O_8_ oxidization–ultraviolet spectrometry	Ultraviolet spectrophotometer Model 752
Ammonia nitrogen (NH_3_–N)	Nessler’s reagent spectrophotometry	Spectrophotometer Model 721
pH	pH meter method	pH meter
temperature (T)	Thermometer method	Thermometer

## Data Availability

The study did not report any data.

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
