# Peer review of "Carbon Reduction and Pollutant Abatement by a Bio–Ecological Combined Process for Rural Sewage"

_ijerph, 2023, doi:10.3390/ijerph20021643_

Round 1

Reviewer 1 Report

Introduction

1. Why rural wastewater has been chose. Literature about its difference from other wastewater.

2. Lines 48-58 not clear. Most probably the mechanism and advantages of bio-ecological wastewater treatment system have been mixed in the write-up. A clear relationship should be focused on this.

3. In 61 addition, the treatment effect of the bio-ecological combined system, especially the eco-62 logical unit, is affected by seasonal changes in the outdoor environment, and the microbial 63 community and structure also changed, which influenced both the microbial metabolism 64 process and the removal efficiency of pollutants” - What is the bio-ecological treatment system referred to in this portion? Is it the same as the experimental set-up used in the present study? Then considering the disadvantages why the same design has been again used.

4. Objective of the study has been poorly explained. A concrete outcome in terms of improving the removal efficiency which would be sustainable should be considered as the aim of the study.

Materials and methods

1. Experimental device or set-up?

2. Why the process is called bio-ecological?Component-wise functional explanation needed.

3. Is there any effluent recirculation system for more purification?

4. Bio-filter needs redesigning.

Results and discussion

1. Component-wise treatment efficiency with explanation needed.

2. Trials needed to increase the treatment efficiency in winter.

3. What about removal potential in other seasons?

4. Experimental trials needed to increase the removal potential by adding carbon sources and by using other macrophytes.

5. A clear mechanistic picture with explanation needed for the ecological processes in oxidation ditch. Which food-chain grazing or detritus more dominant in oxidation pond.

6. No biological parameter has been tested to suggest their contribution potential in removing the pollutants from wastewater.

Conclusion

Too much simple conclusion has been drawn. A lot of work has been done on this kind of treatment system. Novelty is lacking and also the removal potential is less. More follow-up experiments as recommended are needed to increase and optimize the efficiency of the said system.

Author Response

Question 1:Why rural wastewater has been chose.

Response 1:Thank you for kind remind.The economic development of Guangxi is slow, the rural infrastructure is insufficient, and the amount of rural sewage is large and fluctuating, so it is difficult to meet the discharge standard. Therefore, the rural domestic sewage urgently needs to be treated with low cost and high efficiency..

Question 2:Lines 48-58 not clear. Most probably the mechanism and advantages of bio-ecological wastewater treatment system have been mixed in the write-up. A clear relationship should be focused on this.

Response 2:As the reviewer has pointed out, lines 48-58 not clear.We have added the descriptions in the revised manuscript on line 49-60, page 2.

Question 3:“In 61 addition, the treatment effect of the bio-ecological combined system, especially the eco-62 logical unit, is affected by seasonal changes in the outdoor environment, and the microbial 63 community and structure also changed, which influenced both the microbial metabolism 64 process and the removal efficiency of pollutants” - What is the bio-ecological treatment system referred to in this portion? Is it the same as the experimental set-up used in the present study? Then considering the disadvantages why the same design has been again used.

Response 3:Thank you for kind remind.We have added some descriptions in the revised manuscript on line 61, page 2.

Question 4:Experimental device or set-up?

Response 4:Thank you for kind remind.The content of the experimental device has been described in detail in Section 2.1.

Question 5:Why the process is called bio-ecological?Component-wise functional explanation needed.

Response 5:We have added some descriptions in the revised manuscript on line 126-140, page 3.

Question 6:Is there any effluent recirculation system for more purification?

Response 6:Thank you for kind remind.After the rural domestic sewage in this experiment is purified by the biological ecological combined process, the average effluent concentration has reached the Class I A discharge standard of the Discharge Standard of Pollutants for Urban Sewage Treatment Plants (GB 18918-2002), which can be discharged. If the sewage purification device is added, it will cause waste of resources, which does not meet the goal of carbon reduction and emission reduction.

Question 7:What about removal potential in other seasons?

Response 7:Thank you for kind remind.In this experiment, Guangxi domestic sewage is selected in summer and winter because the discharge load of rural domestic sewage is high and the water quantity and quality fluctuate greatly in these two seasons of the year.

Reviewer 2 Report

Abstract; Line 21-25 needs to be clarified. Can you rewrite it?

Line 87; I recommend adding the information on the composition in each layer in biological, subsurface flow constructed wetland, and oxidation pond sections in Figure 1 or show the number and then give the information in the caption of the figure.

-Can you merge Table 1 and Table 2?

- Can you change the data from Table 4 into the text and/or the reference methods that need to be referred to?

-Line 168; What are the four wastewater indicators?

-I recommend changing the axis option from inside tick marks to outside tick marks in Figure 2 and Figure 3, and the error bar (SD) and statistical analysis need to add in the graph.

-Edit the Y-axis of Fig.2a and add the symbols of the removal rate in each figure. The caption of Fig. 2 and figure 3 should add more information on wastewater.

-Line 201-202; Do you have the results to support this discussion?

-Line 212-213; Do you have the results to support this discussion?

-Line 235; Change “pant” to “plant”

-Line 257-258; Do you have the results of DO to support the alternation of aerobic, anoxic, and anaerobic environments in the wetland?

-Line 258; Why do denitrifying bacteria play a role in the system? Do you have the results to support it?

-Figure 4, You should add statistical analysis in the graph.

-I recommend designing the axis by outside tick marks in Figure 6. The error bar (SD) and statistical analysis need to add to the graph, and the scale of the Y-axis should be set in the same number.

Author Response

 Question 1:Line 21-25 needs to be clarified. Can you rewrite it?

Response 1:Thank you for kind remind.We have rewrited it in the revised manuscript on line 21-25, page 1.

Question 2:I recommend adding the information on the composition in each layer in biological, subsurface flow constructed wetland, and oxidation pond sections in Figure 1 or show the number and then give the information in the caption of the figure.

Response 2:Thank you for kind remind.The composition information of each part of the constructed wetland and oxidation pond has been introduced in the experimental equipment. If it is written here again, it will be repeated.

Question 3:Can you merge Table 1 and Table 2?

Response 3:Thank you for kind remind.We have merged Table 1 and Table 2 in the revised manuscript on line 148, page 4.

Question 4:Can you change the data from Table 4 into the text and/or the reference methods that need to be referred to?

Response 4:Thank you for kind remind.We have added some descriptions in the revised manuscript on line 163, page 4.

Question 5:What are the four wastewater indicators?

Response 5:Thank you for kind remind.We have revised it in the revised manuscript on line 170, page 5.

Question 6:I recommend changing the axis option from inside tick marks to outside tick marks in Figure 2 and Figure 3, and the error bar (SD) and statistical analysis need to add in the graph.

Response 6:Thank you for kind remind.We have redrawn it in the revised manuscript on line 180-183, page 5 and line 249-252, page 7.

Question 7:Line 201-202; Do you have the results to support this discussion?

Response 7:Thank you for kind remind.The data supporting this result has been shown in Figure 2.

Question 8:Line 235; Change “pant” to “plant”

Response 8:Thank you for kind remind.We have revised it in the revised manuscript on line 235, page 6.

Question 9:Line 257-258; Do you have the results of DO to support the alternation of aerobic, anoxic, and anaerobic environments in the wetland?

Response 9:Thank you for kind remind.We have added some descriptions in the revised manuscript on line 257-258, page 7.

Question 10:Figure 4, You should add statistical analysis in the graph.

Response 10:Thank you for kind remind.We have added some descriptions in the revised manuscript on line 266, page 8.

Question 11:I recommend designing the axis by outside tick marks in Figure 6. The error bar (SD) and statistical analysis need to add to the graph, and the scale of the Y-axis should be set in the same number.

Response 11:Thank you for kind remind.We have revised it in the revised manuscript on line 358, page 10.

Reviewer 3 Report

General information 

The topic of the review article fits within the scope of the journal, and the special issue. The manuscript elaborates on the real application of biotechnology and engineering in the treatment of real eminent problems. The effect and pissible effects are well elaborated. In general, the manuscript can be accepted after major correction. The major correction is proposed because the authors wrote only one small paragraph at the end of discussion regarding the application of rice straw, corn cobs and common agricultural composted waste as sources of carbon. As I understand, these by-products and wastes were applied at combined bio-ecological process at Guangxi (the remediation facility explained in the manuscript). So why was this not analyzed in detail? Why the efficiency was not calculated? Please address that.

Also, there are some minor corrections needed:

Abstract:

Page 1, line 15. “to construct” please rephrase.

Introduction

Generally, in the phrase bio-ecological combined process the term combined is redundant

Page 2, lines 49-50. Please rephrase this part of the sentence.

Material and methods

Anaerobic pool/ anaerobic tank. What was microbial profile in the anaerobic tank? Was the anaerobic tank used for the production of activated sludge/partial fermentation, or just as a collector tank?

 Page 4, lines 150-151. Dates of collection are a little confusing. Please elaborate.

 Page 4, line 160. Please check Table 4. What is presented and what is mentioned in text?

Results and discussion

Page 5, lines 168-171. Change the order of the sentences, so that the discussion has a more natural flow.

 Page 10, line 359. “havd” please correct.

 Page 10, line 363-364. Please rephrase the sentence. Replace the term “realized”.

References

Please organized the whole reference section by Journal instructions.

Also, the references in text must be separated by “space” from last word in sentence.

 I presume that the authors will continue with the research and the following course of research should be noted.

Author Response

Question 1:Page 1, line 15. “to construct” please rephrase.

Response 1:Thank you for kind remind.We have rephrased it in the revised manuscript on line 15, page 1.

Question 2:Generally, in the phrase bio-ecological combined process the term combined is redundant.

Response 2:Thank you for kind remind.We have revised it in the revised manuscript on line 49, page 1.

Question 3:Anaerobic pool/ anaerobic tank. Was the anaerobic tank used for the production of activated sludge/partial fermentation, or just as a collector tank?

Response 3:Thank you for kind remind.The anaerobic tank was used for the production of activated sludge/partial fermentation.

Question 4:Page 4, lines 150-151. Dates of collection are a little confusing. 

Response 4:Thank you for kind remind.We have revised the date of sewage sample collection in the revised manuscript on line 150, page 3.

Question 5:Page 4, line 160. Please check Table 4. What is presented and what is mentioned in text?

Response 5:Thank you for kind remind.We have added some descriptions in the revised manuscript on line 156-163, page 3.

Question 6:Page 10, line 359. “havd” please correct.

Response 6:Thanks for your valuable suggestion. We have corrected it in the revised manuscript on line 368, page 10.

Question 7:Page 10, line 363-364. Please rephrase the sentence. Replace the term “realized”.

Response 7:Thanks for your valuable suggestion. We have rephrased it in the revised manuscript on line 373, page 10.

Question 8:the references in text must be separated by “space” from last word in sentence.

Response 8:Thanks for your valuable suggestion. We have corrected it in the revised manuscript.

Reviewer 4 Report

The authors propose in the manuscript the use of a combined process composed of a biological filter and a subsurface Flow constructed wetland applied to rural domestic wastewater.

The proposal is interesting, relevant and suitable for IJERPH journal.

Some changes are proposed to make the manuscript clearer.

The introductory section is relevant and has the necessary elements including background and input.

Section 2.1 describes the anaerobic tank and its function, however, it is necessary to include some bibliographic reference to what is described in lines 125-126.

In lines 135-138, 4 areas are described; however, it would be convenient to annex them to the diagram to be more explicit.

Since the study refers to water treatment in a rural area, a general description of the site, location, number of inhabitants, etc., would be useful.

Author Response

Question 1:The introductory section is relevant and has the necessary elements including background and input.

Response 1:Thank you for kind remind.We have introduced the research background in the first section.

Question 2:Section 2.1 describes the anaerobic tank and its function, however, it is necessary to include some bibliographic reference to what is described in lines 125-126.

Response 2:Thanks for your valuable suggestion.We have introduced the anaerobic tank and its functions in detail in 2.1. If we add references to explain it, it will be a bit off the subject.

Question 3:In lines 135-138, 4 areas are described; however, it would be convenient to annex them to the diagram to be more explicit.

Response 3:Thanks for your valuable suggestion.We have already explained that the oxidation pond is divided into four areas. If we introduce it again, it will be a little redundant.

Question 4:Since the study refers to water treatment in a rural area, a general description of the site, location, number of inhabitants, etc., would be useful.

Response 4:Thanks for your valuable suggestion.We have added some descriptions in the revised manuscript on line 67, page 2.

Round 2

Reviewer 1 Report

The study lacks novelty in functioning and mechanism. However, for case specific waste treatment this kind of treatment system can yield some significant results. But the sustainability of the system is questionable since it can operate for two seasons only. What will happen in other seasons in wake of absence of sewage?

Line 51 -original sentence can be maintained 

Reviewer 2 Report

The author has editing followed the comments, and I accepted this manuscript.

Author Response

Thank you very much for your comments.

Reviewer 3 Report

The overall quality of the manuscript has been improved. Thus the manuscript can be accepted for publication.

Author Response

Thank you very much for your comments.